# Midgut development in rat embryos using microcomputed tomography

Marco Ginzel [1,2,4✉], Illya Martynov[2,4], Rainer Haak[3], Martin Lacher[2] & Dietrich Kluth [2]

The development of the mammalian gut was first described more than a century ago. Since then, it has been believed that a series of highly orchestrated developmental processes occur before the intestine achieves its final formation. The key steps include the formation of the umbilicus, the so-called "physiological herniation" of the midgut into the umbilical cord, an intestinal "rotation", and the "return of the gut" into the abdominal cavity. However, this sequence of events is predominantly based on histological sections of dissected embryos, a 2D technique with methodological limitations. For a better understanding of spatial relationships in the embryo, we utilized microcomputed tomography (μCT), a nondestructive 3D imaging method. Here, we show the detailed processes and mechanisms of intestinal development in rat embryos, including the development of the umbilicus, the formation of loops inside the umbilical coelom, and the subsequent shift of these loops into the abdominal cavity. Our 3D datasets of developing intestines will substantially advance the understanding of normal mammalian midgut embryology and offer new possibilities to reveal unknown mechanisms in the pathogenesis of congenital disorders.

---

[1] Department of Neonatology, University Children's Hospital Tuebingen, Tuebingen, Germany. [2] Department of Pediatric Surgery, University of Leipzig, Leipzig, Germany. [3] Department of Cariology, Endodontology and Periodontology, University of Leipzig, Leipzig, Germany. [4]These authors contributed equally: Marco Ginzel, Illya Martynov. ✉email: marco.ginzel@med.uni-tuebingen.de

More than a century ago, Mall[1], and Frazer and Robbins[2] published their fundamental works on gut development based on histological findings of human embryos. They assumed that the normal development of the midgut passes through three key steps: the so-called "physiological herniation", the process of "gut rotation", and the "return" of the intestinal loops from the umbilical coelom into the abdominal cavity. Since then, it has been widely accepted that a limitation of the intra-abdominal space of the embryo accompanied by rapid growth and elongation of the intestine leads to a temporary "physiological herniation" of the gut into the umbilicus (PUH)[3,4] before returning back into the abdominal cavity. During this process, the intestine experiences several rotational steps in a counter-clockwise direction.

Recent studies have addressed these theories of gut embryogenesis. In 2011, our group studied midgut development using scanning electron microscopy (SEM) and questioned the hypothesis of rotation[5]. In the same year, Savin et al. developed a model of gut looping in chicken embryos, suggesting the importance of mesenteric elasticity for intestinal loop formation[6]. Soffers et al. (2015) addressed the development of the whole intestine and concluded that the intestine does not "rotate" but "slides" from the umbilical coelom into the abdominal cavity[7]. Recently, Nagata et al. (2019) studied the return of intestinal loops after PUH, proposing an alternative "wrapped model" instead of the classical "rope model"[8].

However, many aspects important for embryonic gut development are still not fully addressed or have conflicting evidence. For instance, detailed studies on the formation of the umbilicus are scarce, which limits our understanding of the so-called PUH. Furthermore, the exact mechanism of the "midgut return" is yet unknown. Although discussions on midgut development are believed to be largely settled, detailed and precise morphological investigations of these processes are still missing.

Thus, we restudied midgut development in rat embryos from embryonic day 10 (ED 10) until one day prior to birth (ED 21) using microcomputed tomography (μCT) as a tool for detailed embryonic studies. This technique is an attractive option for morphological and morphometrical analysis of rodent embryos, which enables precise visualization and accurate measurement of 3D structures without the need for embryo dissection[9–11].

Here we show the complete development of the rat midgut and of the umbilicus, clarifying unknown mechanisms and providing information about the impact of the elasticity of blood vessels on midgut development.

## Results

### The preumbilical ring and its relationship to the surrounding embryonic and extraembryonic membranes (ED 10 – ED 11).
We started our morphological investigation at ED 10, a time point at which the first intestinal structures including the foregut and hindgut diverticulum had formed (Fig. 1a). At ED 10, the gut epithelium (embryonic endoderm) of the unfolded rat embryo extended ventrally and was in continuity with the wall of the yolk sac, which yet showed no signs of vitelline vessels. The allantois (precursor of the body stalk in rodents) emerged from the most caudal part of the embryo and extended dorsally into the extraembryonic coelom. Its tip pointed to the chorion of the ectoplacental cone area. At this time point, umbilical vessels inside the allantois were not detectable (Fig. 1a). An endodermal hindgut diverticulum (the so-called "allantoic vesicle") projecting into the mesenchyme of the allantois was not found at this stage in contrast to human embryos.

At ED 10, the amniotic cavity was positioned dorsally to the embryo. Its membranes (extraembryonic ectoderm and mesoderm) were in continuity with the embryonic ectoderm and mesoderm of the lateral body wall of the embryo. In this transition zone, a distinct epithelial fold was seen, forming a ring-like structure, which marked the lateral end of the body wall. We defined this structure as the preumbilical ring (Fig. 1a, b).

At this stage, the lateral plate mesoderm of the embryo was already split into two layers, the somatic and splanchnic mesoderm[12,13]. The splanchnic mesoderm was closely connected to the embryonic and extraembryonic (yolk sac) endoderm (splanchnopleura), while the somatic mesoderm was attached to the embryonic and extraembryonic (amnion) ectoderm (somatopleura).

Between ED 10 and ED 11, the embryo went through the processes of body turning and body folding, as previously demonstrated by Kaufmann[14]. Hence, the cranial part of the amniotic membrane (head amnion) was in contact with the caudal part (tail amnion) (Fig. 1c, d). Furthermore, the plane of the preumbilical ring as well as the allantois/body stalk, which was initially positioned dorsally (Fig. 1a), was now located ventrally (Fig. 1c). The allantois/body stalk had now established firm contact with the chorion of the ectoplacental cone area and contained umbilical vessels. It was connected to the somatopleura of the embryo caudally and from the right in an extraperitoneal fashion. In contrast, the developing vitelline vessels intraperitoneally entered the splanchnopleura of the embryo centrally and from the left (Fig. 1c, d, 2b(i), and Supplementary Video 1). After accessing the embryo, the umbilical vein was part of the developing ventrolateral body wall (somatopleura) where this vein separated into a right- and a left-sided branch. The right-sided branch formed the ridge of the developing right ventrolateral body wall on its way to the right sinus horn of the heart (Figs. 2b(ii) and 3a). The left-sided branch crossed the body midline from right to left, formed the ridge of the developing left ventrolateral body wall and finally entered the left sinus horn. Thus, at ED 11, the preumbilical ring reached a new developmental configuration, which we refer to as the "intermediate umbilical ring". This ring is formed by the paired umbilical veins (right and left) and the sinus venosus cranially (Fig. 2b(ii)). At this stage, both the umbilical and the vitelline vessels were surrounded by amniotic membranes (extraembryonic somatic mesoderm and ectoderm), giving the appearance of two funnels (the umbilical funnel and the vitelline funnel) (Figs. 2b(i), c, 3a, and Supplementary Video 2). The vitelline funnel (mainly covered by head amnion), containing the vitelline vessels, the ductus omphaloentericus and the midgut, formed the "vitelline compartment" of the umbilicus (Supplementary Figs. 1, 2). This compartment allowed free communication between the intra- and extraembryonic coelom of the embryo until late ED 12 (Fig. 2c and Supplementary Video 3). The umbilical funnel with its umbilical vessels shaped the "umbilical compartment" (half covered by head and tail amnion). In contrast to the vitelline funnel, the umbilical funnel ended at the body wall. Thus, the umbilical and vitelline compartments were clearly separated at ED 11. The axis between these compartments was oriented from left-cranial to right-caudal (Fig. 3a).

### Development of the intermediate umbilical ring and formation of the definitive umbilicus (ED 11 – ED 13).
Beginning on ED 11, an asymmetric transformation of the paired umbilical veins took place. The downstream portion of the left umbilical vein lost its connection to the sinus venosus of the heart and disintegrated in this area. Simultaneously, the upstream portion of the left umbilical vein remained part of the developing ventrolateral body wall but established a new connection to the hepatic blood vessels via the midportion of the septum transversum. During further

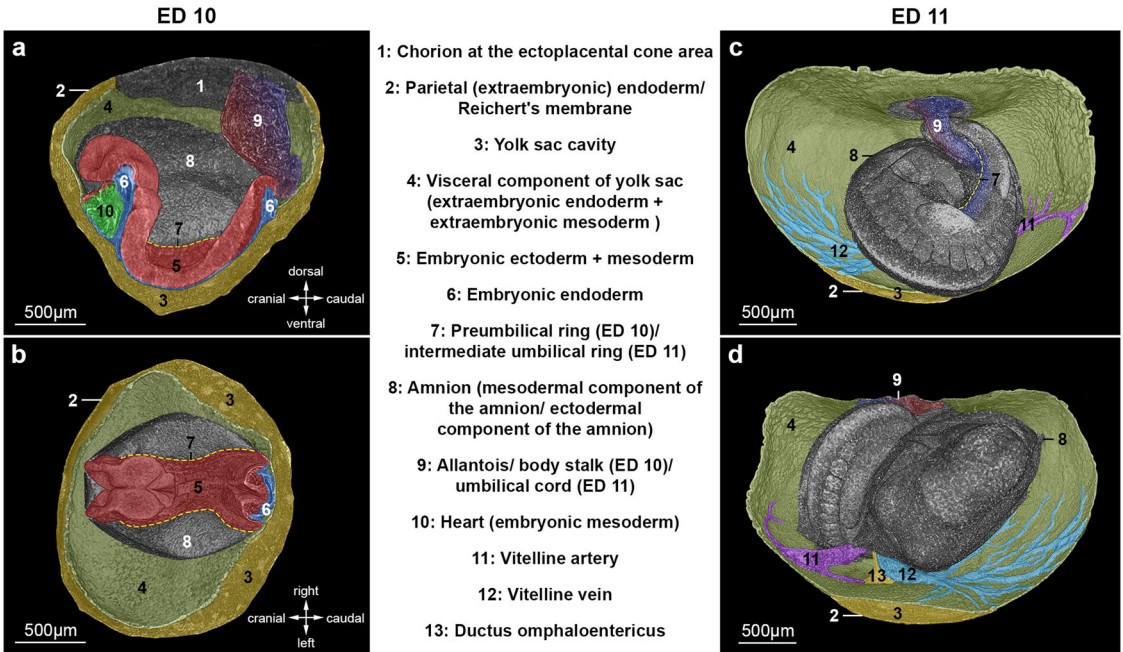

**Fig. 1 Normal development from the pre- to intermediate umbilical ring and the surrounding embryonic and extraembryonic tissues.** Morphological comparison of rat embryos at ED 10 (before body turning and body folding) and ED 11 (after body turning and body folding). **a** ED 10, right lateral view after virtual sagittal sectioning of the embryo. **b** Lateral view after sagittal sectioning with the head on the left. The preumbilical ring represents the border between the embryonic and extraembryonic ectoderm and is located dorsally. The allantois/body stalk is located at the most caudal part of the embryo and is connected to the chorion (developing placenta). Vitelline vessels have not yet been formed. **c** ED 11, view on allantois/body stalk. The embryo is turned and folded, while the orientation of the allantois/body stalk is unchanged. Vitelline vessels appear on the opposing side of the allantois/ body stalk. The midgut is still connected to the yolk sac through the ductus omphaloentericus. The preumbilical ring moves ventrally with the umbilical veins to form new borders. **d** ED 11, view on the vitelline vessels and the ductus omphaloentericus.

development, the right umbilical vein, which formed the edge of the lateral body wall until early ED 12, disintegrated completely into smaller vessels at late ED 12 (Fig. 2c). The development of the left umbilical vein differed from the right side, by increasing in diameter. Simultaneously, through the complete disintegration of the right umbilical vein, the left umbilical vein loses its connection to the right vein and thus gets more and more separated from the lateral and caudal body wall. As a result, the left umbilical vein becomes an integral part of the definitive umbilicus at ED 13 (Fig. 2b, c).

In connection with the general changes in the external shape of the embryo (body rotation, body folding), the umbilical and vitelline vessels moved closer to each other in craniocaudal and mediolateral directions (Fig. 2b(ii)). This process of approximation formed the umbilicus as a discrete structure (Fig. 3b), in contrast to previous assumptions (Fig. 3c). However, both compartments remained separated from each other (Fig. 3b, d). The umbilical ring was no longer formed by the umbilical vessels. Instead, the developing ventrolateral body walls (somatopleura) now represented the boundaries of the definitive umbilical ring.

This final developmental stage was reached at ED 13. Here, the body stalk (umbilical vessels and the surrounding mesenchyme) is clearly discernable as an entity separated from the vitelline compartment. This compartment, which in the early phase of development allowed free communication between the intra- and extraembryonic coelom of the embryo, closed ventrally between late ED 12 and ED 13, thus forming a sac-like structure that now established the extraembryonic coelom of the "umbilicus" (Fig. 3b, d).

**Growth dynamics during the formation of the first intestinal loop (ED 11 – ED 13).** Beginning on ED 11, a sickle-shaped midgut had formed and was located in the center of the

developing abdominal cavity (Fig. 2b(ii), 4a). The initially large connection between the lumina of the gut and yolk sac had been reduced to form a narrow yolk sac canal (the vitelline duct), which later diminished completely (Figs. 2c and 4a). The vitelline artery, the precursor of the superior mesenteric artery (SMA), was attached to the midgut wall and entered the intraperitoneal cavity of the embryo centrally and from its left side. At ED 12, the midgut had formed its first loop in the area where the vitelline artery left the embryo. This loop was thus orientated to the left side (Fig. 2b). Furthermore, we did not see a limitation of space in the abdominal cavity when the first midgut loop developed (Supplementary Fig. 3).

The anlage of the cecum emerged in the early stages of ED 12 as a thickening of the caudal portion of the intestinal loop inside the developing abdominal cavity. With further development, the growing intestine led to an increase in loop length, thereby shifting the cecum from caudal to cranial. Finally, it reached a location within the vitelline compartment of the umbilicus outside the borders of the umbilical ring. The cecum maintained its position inside the vitelline compartment close to the border of the umbilical ring until the cecum and midgut entered the abdominal cavity at ED 17.

Through the elongation of the colon, the colonic limb of the midgut loop was forced cranially. Due to the position of the left vitelline vein inside the embryo the duodenal flexure was located to the right side of the abdomen. The cranial shift of the colonic flexure caused its positioning to the left of the duodenal–jejunal junction (Fig. 4a). This developmental process was interpreted in the past as a first step of midgut rotation, caused by an unknown external force. However, we saw that this process was rather the result of a convergence of the colonic and duodenal intestinal limb, induced by growth of the intestinal loop and thickening of the left umbilical vein (Supplementary Videos 4 and 5).

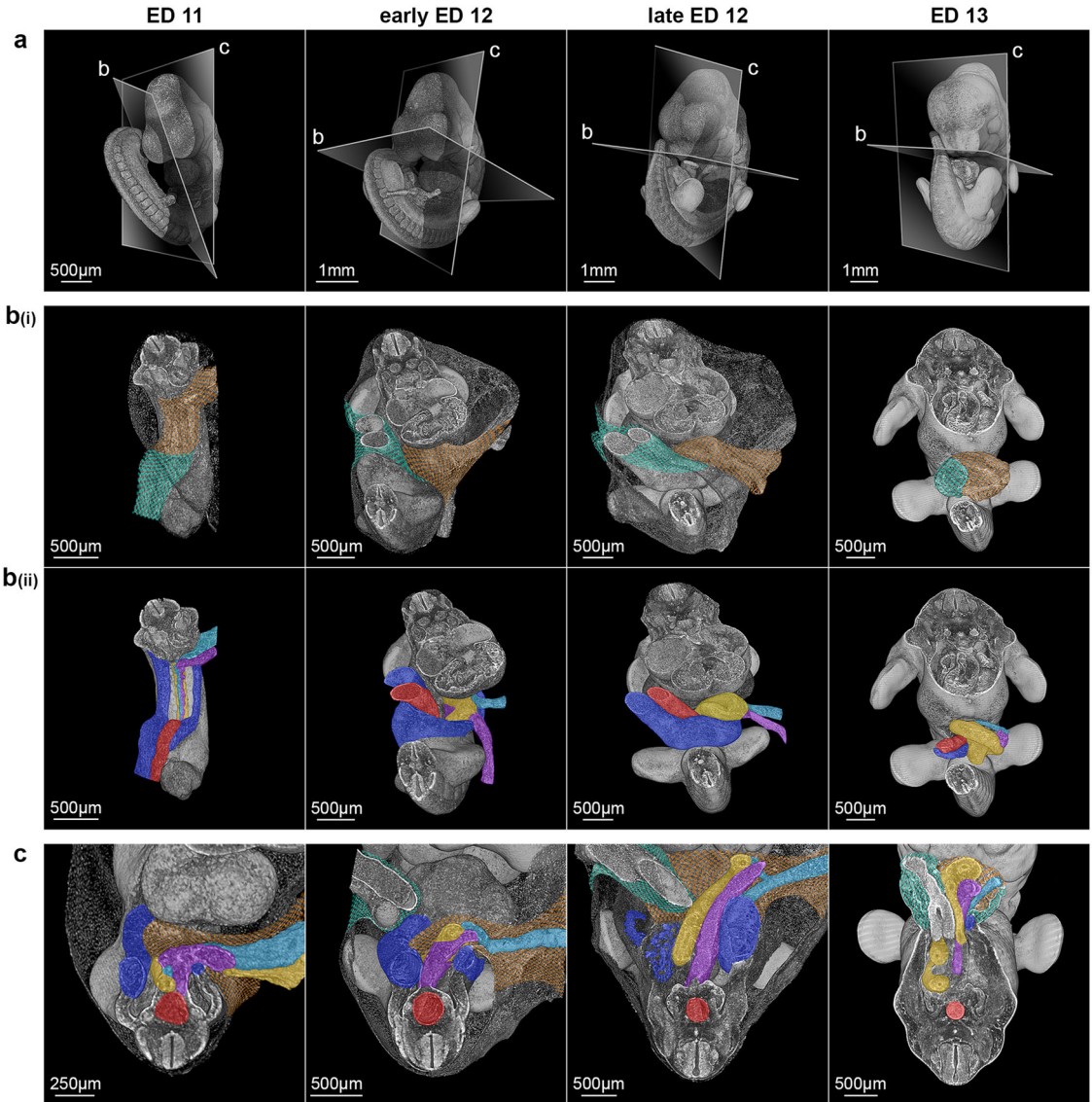

**Fig. 2 Formation of the definitive umbilicus. a** Representative reconstructions of rat embryos from ED 11 to ED 13 showing embryonic torsion and sectional planes of (**b**) and (**c**). **b(i)** Ventral view of the amnion-covered umbilical region. Amniotic membranes form two compartments at ED 11 (umbilical compartment: turquoise net; vitelline compartment: orange net), which aggregate during development to ED 13. **b(ii)** Ventral view after virtual removal of the amniotic membrane. Vessels and the intestine are colorized (intestine/yolk sac channel: yellow; umbilical artery/aorta: red; umbilical veins: blue; vitelline artery: purple; vitelline vein: turquoise). **c** Transversal section of the embryo in the area of the developing umbilicus. The asymmetric development of the right and left umbilical veins is shown.

**Loop and vessel arrangement of the small intestine within the extraembryonic coelom (ED 13 – ED 16).** From ED 13 to ED 16, further intestinal growth occurred predominantly inside the extraembryonic coelom of the umbilicus by the formation of various loops. Loop formation inside the extraembryonic coelom started between ED 13 and ED 14 and was only observed in the region of the small intestine (Fig. 4a, b). Intestinal growth caused the tip of the first loop to bend in the caudal direction, resulting in the formation of two additional loops (Fig. 4b). This formation was seen uniformly in all investigated rat embryos of this age group and could also be observed in human embryos of comparable age[15], thus representing the last stage of stereotypical development.

While the number of loops was the same in similar developmental stages in different rats, their 3D orientation differed. Despite this finding, a specific fixed arrangement of blood vessels supplying the small intestine could be observed. We

could clearly identify three different bundles of vessels originating from and limited to the caudal surface of the SMA at ED 16, disposed like a spiral staircase (Fig. 4b). These vessels separately supplied three clusters of intestinal loops. Counting the number of vessels allowed the assignment of clusters at ED 15 and even at ED 14 retrospectively. These clusters remained distinguishable as long as they reached the vitelline compartment of the umbilicus. However, during the intestinal shift from the vitelline compartment of the umbilicus into the abdominal cavity occurring at ED 17, the initial arrangement of the clusters vanished. After that stage, the loops of each cluster could still be identified by the number of their supplying vessels.

**Staged shift of the midgut from the vitelline compartment of the umbilicus to the abdominal cavity (ED 17).** At ED 17, the shift (or so-called "return") of the midgut from the

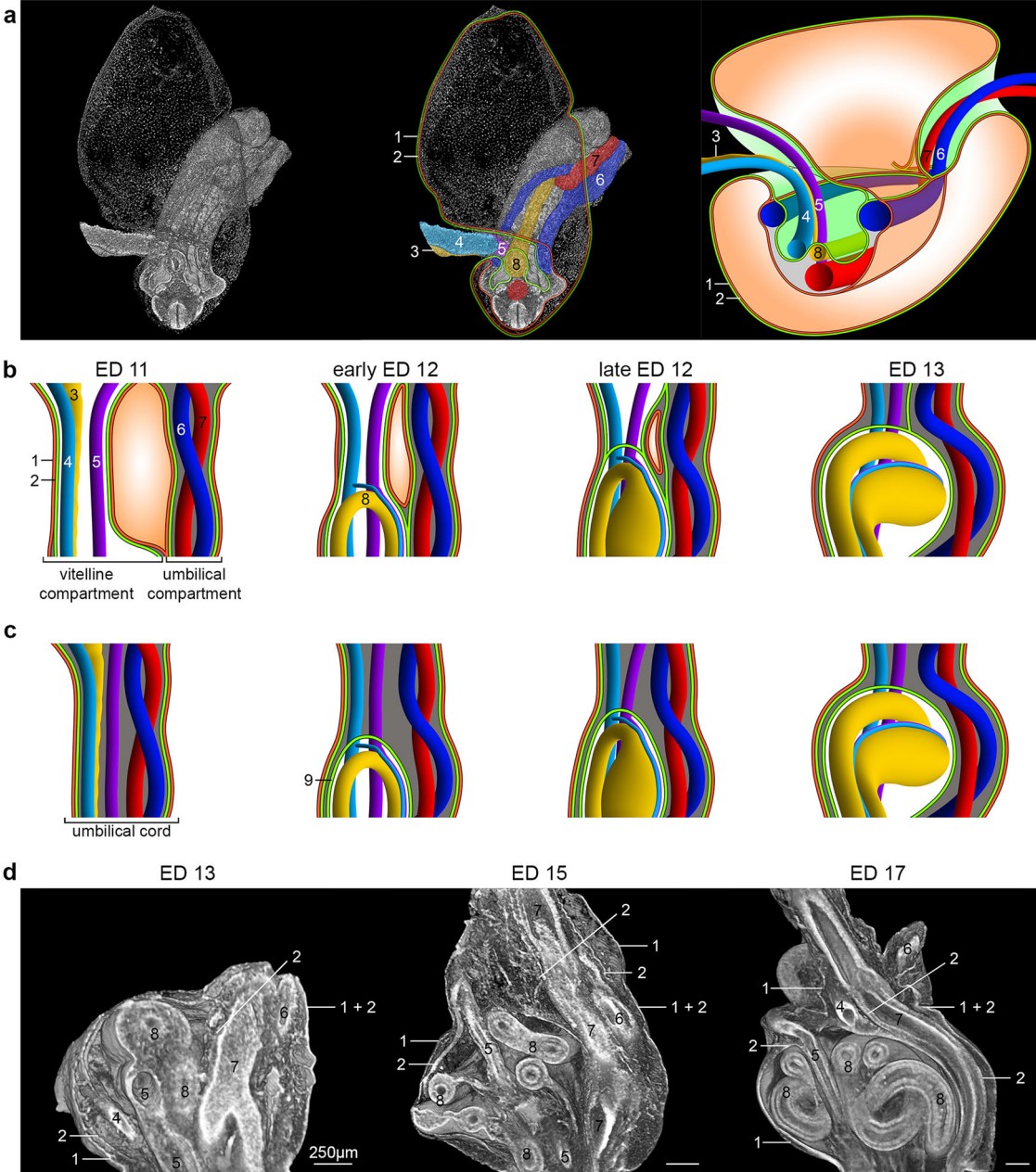

**Fig. 3 Comparison of extraembryonic coelom development during umbilicus formation. a** Steps from a reconstructed ED 11 embryo with a view of the abdominal area (the head was virtually removed) to a schematic illustration. **b** Our concept of extraembryonic coelom development. Here, the umbilicus is not represented by the umbilical cord only but by two compartments formed by the amnion. Initially, these are clearly separated at ED 11 but then aggregate to form the definitive umbilicus until ED 13. In this process, the vitelline compartment, which is continuous between the intra- and extraembryonic coelom of the embryo until late ED 12, is sealed ventrally to form the coelomic sac, covering the extraembryonic midgut loops. **c** Common schematic illustration of extraembryonic coelom development[4,16]. According to these drawings, the extraembryonic coelom appears in the context of the "herniation of the midgut", which occurs inside the umbilical cord. **d** Representative reconstructions of embryos from ED 13 to ED 17 showing the structures of the definitive umbilicus. 1: Ectodermal component of the amnion, 2: Mesodermal component of the amnion, 3: Ductus omphaloentericus, 4: Vitelline vein, 5: Vitelline artery, 6: Umbilical vein, 7: Umbilical artery, 8: Midgut, and 9: Mesoderm surrounding the gut, forming the extraembryonic coelom.

extraembryonic coelom of the umbilicus into the abdominal cavity took place. For a better understanding of this process, the initial situation of the intestine and its supplying blood vessels at ED 16 will be described first. At this stage, the extraembryonic coelom of the umbilicus was densely filled with intestinal loops, the cecum, and a small fraction of the colon. At the area of the umbilical ring, the supplying blood vessels, originating from the SMA and SMV, showed some bending towards the intestinal connection sites (Fig. 5a). This bending implied minimal tension

on the vessels at that time. Starting at ED 17, morphological changes took place, resulting in an increased stretch on the vessels supplying the intestine. As the SMA is anchored to the aorta, the growing abdominal scope accompanied by an increased distance between the aorta and the ventral abdominal wall led to this stretch (Supplementary Fig. 4a, b). Subsequently, a significant widening of the umbilical ring could be observed ($p < 0.001$) by the stretched SMA branches (Figs. 5b, c and Supplementary Fig. 4c). Finally, the capacity of the SMA and its branches to

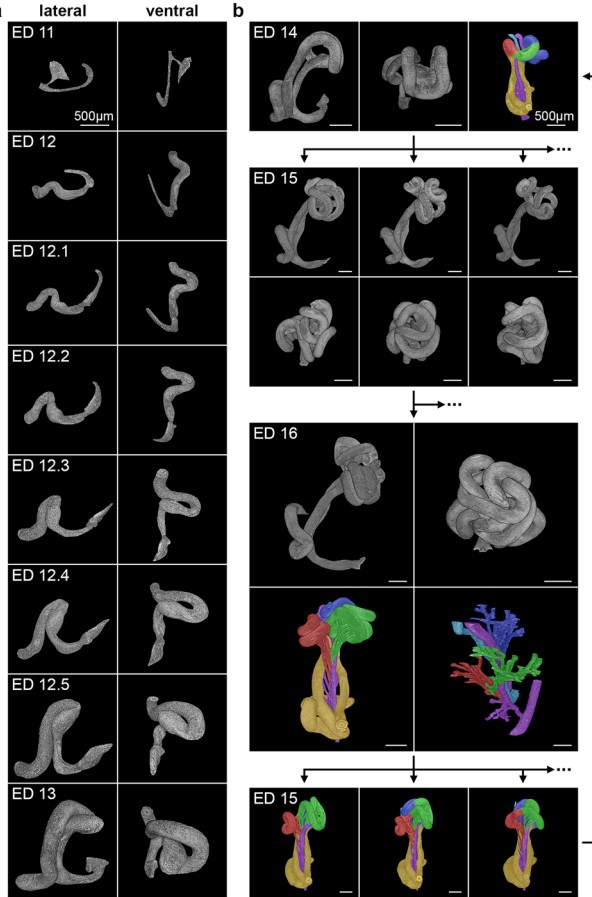

**Fig. 4 Intestinal growth patterns from ED 11 to ED 16. a** Scaled development of the first intestinal loop. The ductus omphaloentericus disappears between ED 11 and ED 12, and the first intestinal loop is formed in that area. At ED 12.1, the cecum appears and allows separation between the small and large intestines. The picture sequence from ED 12.2 to ED 13 shows the convergence of the duodenal and colonic limb, resulting in a side-by-side position of the colon and the intestine, mimicking a rotation of 90°. This growth pattern was observed in all animals in these age groups and therefore represents a conserved growth pattern. **b** ED 14 represents the last developmental stage with this stereotypical pattern. Further intestinal growth leads to a semi-random arrangement of additional emerging loops, as shown in ED 15. This semi-random loop growth continues. Evaluation of the supplying vessel system on ED 16 unveiled an underlying pattern, which allows the understanding of this semi-random loop arrangement. Hence, structures, such as vessels and loop clusters, are colorized (duodenum + colon: yellow, 1st cluster: red, 2nd cluster: green, 3rd cluster: blue, vitelline artery: purple, vitelline vein: turquoise). This pattern in turn can be traced back to ED 15 and eventually to ED 14.

length of the SMA increased from ED 16 to ED 17.0 ($p = 0.02$) and decreased significantly after the shift ($p = 0.003$, Fig. 5d). The mean diameter of the SMA branches showed a stagnation from ED 16 to ED 17 and a significant increase after the shift ($p < 0.001$, Supplementary Fig. 4d).

The shift of the cecum occurred during ED 17.3. However, the terminal ileum was the last loop to enter the abdominal cavity. The most ventral part of the colon shifted passively, entrained by the shifting cecum during the final relocation steps, as the colon as well as its ridge vessel remained relaxed during the shifting process (Supplementary Fig. 5). Finally, the entire intestine found its place inside the abdominal cavity, competing for space with other organs (i.e., liver, urinary bladder). With the final loop entering the abdomen, the umbilical ring started to close, leaving a gap for the remaining umbilical and vitelline vessels.

**Morphological overview and overall growth dynamics (ED 14 – ED 21).** We finally describe the overall intestinal growth, beginning from the last stereotypical morphological stage (ED 14) until one day prior to birth (ED 21). Throughout development, the formation of intestinal loops proceeded, but with different positioning of these loops in each animal. Therefore, the individual 3D patterns of the small intestine increased with age. In contrast, the morphology of the colon and the duodenum in the central area remained preserved. The colon was positioned dorsally and showed some curving with increasing age, but the overall shape remained constant. The duodenum showed a "C" shaped form in all age groups, excluding a malrotation of the gut in wild-type rats examined in this study (Fig. 6a).

For the final analysis, we evaluated the morphometric values of four animals per age group from ED 14 to ED 21. Throughout the investigated period, the whole intestine (duodenum, small intestine, cecum, and colon) showed exponential growth in volume (Fig. 6b), diameter, and length (Supplementary Fig. 6). Furthermore, by comparing the growth patterns of different gut segments based on the volumetric distribution on a percentage basis, we found that the small intestine (jejunum and ileum) showed the highest growth rates. The duodenum showed constant growth, whereas the cecum and colon grew more slowly (Fig. 6c).

## Discussion

The majority of previous studies on midgut development are based on human embryos and aim to explain the embryological mechanisms through clinically observed malformations, in particular malrotation and body wall abnormalities[4,15–19]. Consequently, assumptions and conclusions deriving from pathological observations were used to explain normal midgut development[3,20]. In our study we applied micro-CT-scan technology on rat embryos to document the morphological changes of normal midgut development and its surrounding structures. This technique allows a 3D reconstruction of undissected specimens and the measurement of volumes and lengths, thus adding morphometric data to morphological observations. While our general findings confirm previously described stages of midgut development, we want to address the following developmental steps in more detail: 1. The development of the umbilicus, 2. the development of the first midgut loop and its relationship to vitelline structures, and 3. cluster formation and staged shift of the midgut.

The development of the umbilicus starts with the formation of the umbilical ring. According to Hartwig et al., the umbilical ring first appears as the transition zone between the embryonic and extraembryonic ectoderm of the embryo, but no further subdivisions were made[17,18]. Although we found the same umbilical

stretch further was depleted, forcing the intestinal loops to shift. After the shift, the morphology of the SMA and its branches showed a relaxed configuration (Fig. 5a). This shift was highly organized and took place in a clear order. The loops of the first cluster, which were connected to the most proximal SMA branches were forced into the abdominal cavity first. Thus, we differentiated five distinct stages, from ED 17.0 (before the shift), through ED 17.1 – ED 17.3 (shift of each cluster), to ED 17.4 (the midgut is located completely inside the abdominal cavity).

To assess the assumed stretching of the supplying midgut vessels we analyzed the diameters of the SMA and its branches. In addition, we measured the volume of the SMA to calculate its length. Our morphometric data showed that the diameter of the SMA was smaller dorsally compared to ventral. The calculated

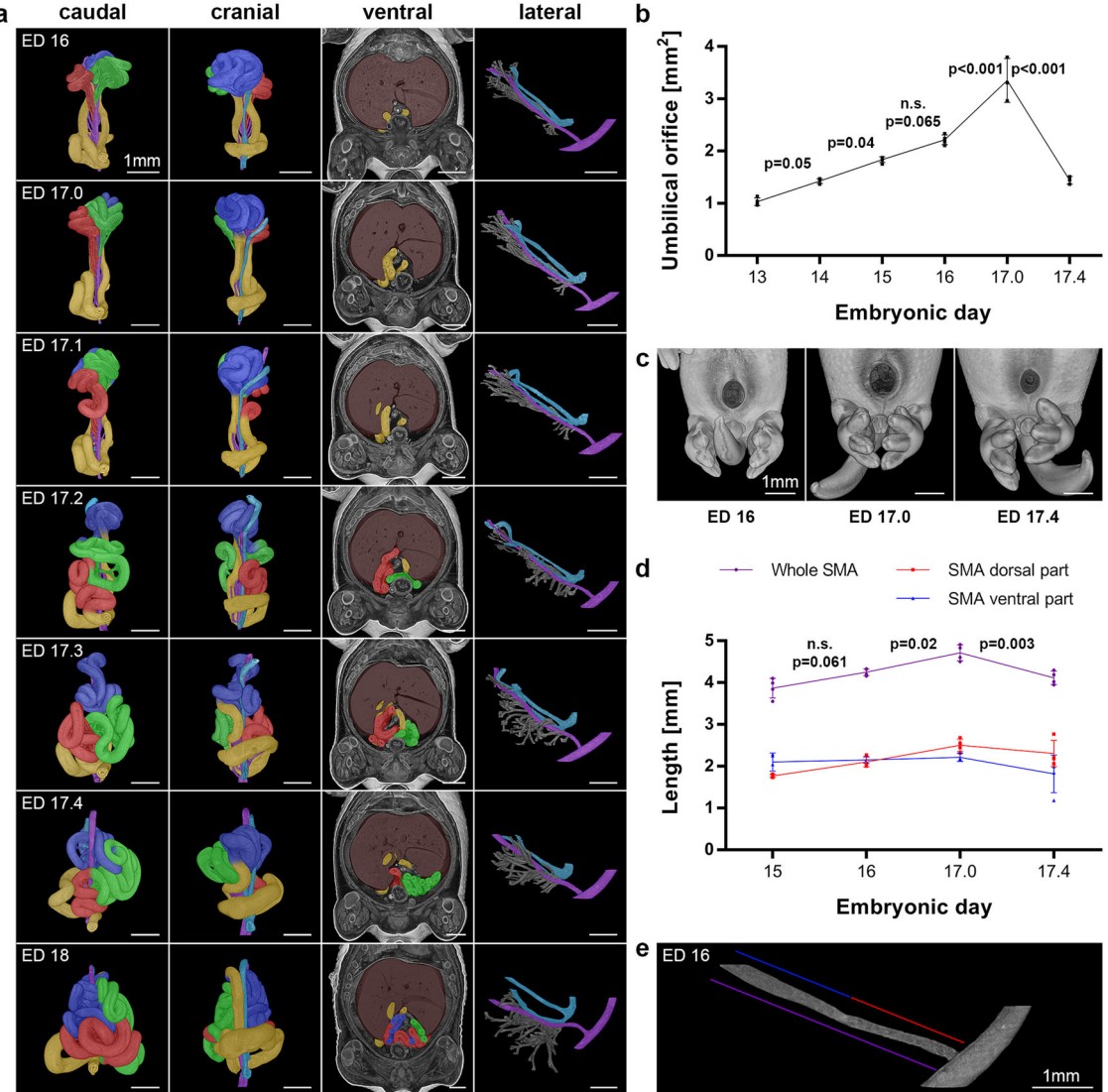

**Fig. 5 Intestinal shift from the umbilicus to the abdomen. a** Single steps of the midgut shift at ED 17 (ED 17.0: before the shift, ED 17.1: shift of the 1st cluster, ED 17.2: shift of the 2nd cluster, ED 17.3: shift of the 3rd cluster, ED 17.4: accomplished shift) are shown, including the previous and subsequent stages at ED 16 and ED 18. Colorization of intestinal clusters and vessels is shown in Fig. 4. The intestinal morphology is shown from a caudal and cranial view. The ventral view shows the intestine in the abdominal cavity with the surrounding liver (brown). The isolated vessel system is shown from the lateral side. **b** Morphometry and morphology of the umbilical ring during the shifting process. The area of the umbilical orifice was calculated, and changes over time are shown as single data points with SD ($n = 3$–4 animals). Statistical test: One-way ANOVA with a Bonferroni post-hoc test, two-sided. Effect sizes in Cohen´s $f$: ED 13 – ED 14: 0.28, ED 14 – ED 15: 0.29, ED 15 – ED 16: 0.27, ED 16 – ED 17.0: 0.59, ED 17.0 – ED 17.4: 1.06. **c** Representative reconstructions of embryos with a virtual removed umbilicus. **d** The calculated lengths of the SMA over time are shown as single data points with SD ($n = 4$ animals). Statistical test: One-way ANOVA with Bonferroni post-hoc test, two-sided. Effect sizes in Cohen´s $f$: ED 15 – ED 16: 0.1, ED 16 – ED 17.0: 0.12, ED 17.0- ED 17.4: 0.16. **e** Dorsal (red) and ventral parts (blue) of the SMA were analyzed independently. Whole SMA length is indicated in purple.

structure in rat embryos at ED 10, we rather refer to it as pre-umbilical ring because we observed two more distinct morphological steps, the intermediate- and definitive umbilical ring. The intermediate umbilical ring (ED11 to ED12) is represented by the now formed paired umbilical veins positioned at the ventral ridge of the lateral body wall running from caudal to cranial. The morphology of this ring is further changed by the disintegration of the right umbilical vein at late ED 12 releasing the left umbilical vein which then folds away from the lateral body wall to become a part of the definitive umbilicus at ED 13. At the end of this transition, the definitive umbilical ring has formed as a central structure with the umbilical vein inside the ring at its left border. Thus, we show that the varying vessel morphology has a direct impact on the development of the definitive umbilicus. In

most publications, vessel development is mainly mentioned to explain congenital malformations such as gastroschisis. This abdominal wall defect is explained by an abnormal disappearance of the right umbilical vein causing a defect in the right lateral body wall[21–23].

In that context, we need to address that the turning of the embryo between ED 10 and ED 11 causes a twist of the amniotic cavity, resulting in the formation of two "funnel-like" structures where the umbilical and vitelline vessels enter the embryo (umbilical and vitelline funnel). In contrast to this, previous reports present drawings, which suggest that the yolk sac and the umbilical cord are located in a single compartment surrounded by the amniotic membrane[4,17,22,24,25]. Furthermore, some publications indicate that the hernial sac at the base of the umbilical

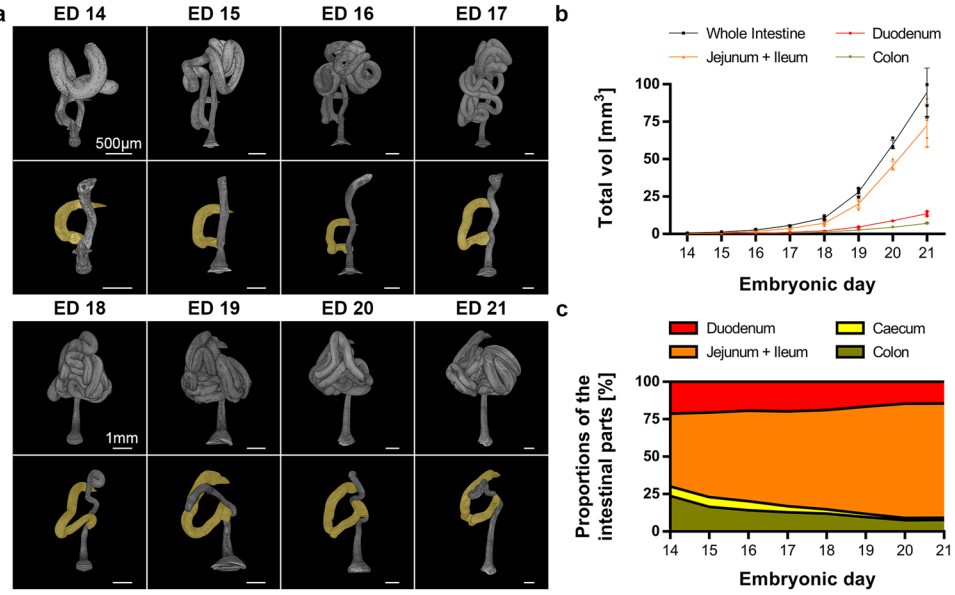

**Fig. 6 Intestinal growth from ED 14 to ED 21. a** Representative reconstructions of the whole intestine (upper row of each age) and of the separated duodenum (yellow) and colon (lower row of each age) are shown. While the small intestine shows an inconsistent looping, the morphology of the colon and the duodenum is preserved. The duodenal "C" is present in all age groups. **b** Volumes of all intestinal segments are shown over time as single data points with SD (*n* = 4 animals). **c** The proportions of intestinal segments are shown over time.

cord is created by the herniating midgut itself[4,16,26]. Our study in rat embryos showed, that the mesoderm of the later extra-embryonic coelom of the umbilicus is already present at ED 11 caused by the turning of the embryo. This vitelline funnel is in continuity from extraembryonic to the intraembryonic coelom at ED 12, the time, at which the first midgut loop appears. This continuity is interrupted in the later phase of development at late ED 12 by the ventral closure of this funnel, resulting in the image of a "hernial sac". Thus, we hypothesize, that in case of gastroschisis a closure of this funnel did not take place.

The formation of the first midgut loop is traditionally explained by a lengthening of the midgut, which exceeds the general growth of the embryo. Furthermore, the growth of the liver should reduce the available intra-abdominal space, which prevents the positioning of the midgut loops inside the abdominal cavity. Therefore, the first midgut loop is forced into the umbilical cord, the so-called PUH[27,28]. However, we do not see a reduction of intra-abdominal space caused by the developing liver (Supplementary Fig. 3). This finding is supported by Kanahshi et al., who studied early gut development in human embryos with liver hypogenesis or agenesis[29]. They noted that a decreased liver volume was associated with a decreased abdominal volume, while total intestinal length, extraembryonic intestinal length, and its ratio were nearly similar in all embryos studied, independent from the size of the liver. Thus, the authors challenged the mechanism of PUH caused by rapid normal liver growth. However, their study gave no hint why midgut loops are found in the extraembryonic coelom. In theory, looping of the gut should be possible anywhere in the area of the developing midgut. However, we noticed a defined special relationship between the vitelline structures (vitelline vessels and ductus omphaloentericus) and the area of the developing first midgut loop through their attachment points to the midgut wall. We assume that the orientation of the tip of the loop outside the abdominal cavity and to the left side of the embryo is a result of a synchronized growth of the vitelline vessels and the attached midgut. Furthermore, the increased distance between the midgut and the aorta can explain the formation of the midgut mesentery. In the whole process of the formation of the first midgut loop, the connections of the vitelline

vessels to the yolk sac seem to serve as an anchoring point, which directs the tip of the midgut loop outside and to the left of the embryo (Supplementary Fig. 7). The turning of the embryo might contribute to a pull in the direction of the yolk sac.

After the appearance of this first midgut loop, we noticed an approximation of the duodenal and colonic limbs resulting in the formation of a duodenal and the colonic flexure, which converge in a side-to-side fashion. The resulting morphology is often described as midgut "rotation", which should take place after the formation of the midgut loop by unknown external forces[2,30]. Our results are supported by others who believe that this morphology is not caused by "rotation" but by convergence of the duodenal and colonic flexures caused by differential growth of these structures[5,31].

The development of intestinal loops inside the extraembryonic coelom of the umbilicus is the most prominent feature in early midgut development. According to Savin et al., the length of the midgut vessels combined with intestinal growth induces the formation of loops in chicken embryos[6]. Our observations in rat embryos are in agreement with their findings, as the supplying midgut vessels appear to be limited in their length, forcing the growing midgut to form loops. However, while chickens have no limitation of space within their extraembryonic coelom for the developing midgut, rat embryos have a confined umbilical compartment (extraembryonic coelom)[5]. These physical borders have an impact on the arrangement of midgut loops. As shown in Fig. 4, the loop orientation was variable from ED15 onwards in different specimens. Thus, we focused on the vessel patterns and their relationship to the developing midgut loops. We could clearly identify three bundles of vessels on ED 16, each supplying a group of midgut loops, we referred to as a cluster. We assume, that this formation had its first visible origin at ED 14. However, in this stage we were not able to identify the accompanying vessels due to technical limitations. These clusters resemble basically the previously described 2nd to 4th secondary loops by Soffers et al.[7]. However, the subdivision by Soffers et al. is based on intestinal loop organization in a chronological order, while the three extraembryonic secondary loops are connected to only two mesenteric leafs. Our classification uses vessel morphology, which

allows a clear identification of each cluster, based on the three bundles of vessels. Furthermore, we do not subdivide these clusters further into tertiary and quarterly loops, as proposed by Soffers et al. because this subdivision was not supported by the observed vessel pattern. However, the vessel configuration seems to be important for the shift of the midgut into the abdominal cavity on ED 17.

This shift itself was first described by Mall[1] and later by Frazier and Robbins[2]. Mall proposed in 1899, that the midgut loops inside the extraembryonic coelom gets "sucked back" into the abdominal cavity. He suggested, that an extension of the abdominal cavity resulted in a kind of negative pressure[1]. In 1915, Frazer and Robbins modified the mechanism for the shift, indicating that amniotic pressure on the umbilical sac might contribute to the "suck back" mechanism. The shift itself is described to take place in a specific order; the intestinal segments shift from proximal to distal and not end block, as a narrow umbilical orifice would make this process unlikely[2]. Based on results from Soffers et al., Nagata et al. proposed a mechanism, called "wrapped model". In short, this theory describes a process in which the intestinal loops are wrapped into the abdominal cavity as the height of the umbilical ring increases ventrally[7,8].

Interpreting our results, we propose the following mechanism: The shift is triggered by the increasing abdominal scope. In effect, this scope increases the distance between the aorta and the umbilical orifice. Consequently, the midgut loops inside the extraembryonic coelom are constantly pushed to ventral, as the umbilical orifice is too small for the shift to occur until ED 17. From ED 16 to ED 17 the SMA branches get stretched in dorso-ventral direction forming a cone-like structure which subsequently widens the umbilical orifice before the shift takes place (Supplementary Fig. 3).

The shift itself takes place in a highly organized fashion. It starts with the proximal loops of the 1st cluster and ends with the most distal loops of the 3rd cluster. The cause for this order is the arrangement of the supplying vessels to the midgut loops. Their outlets from the SMA are orientated in a dorso-ventral manner with the most dorsal SMA branches supplying the proximal midgut loop and the most ventral branches supplying the distal loop. In detail, during the ongoing ventral shift of the umbilical orifice and the stretching of all SMA branches, the supplying vessels to the most proximal midgut loop are the first to lose their elasticity and thus become rigid. When this happens, this loop shifts through the ventrally moving umbilical orifice into the abdominal cavity. Afterwards, the vessels supplying this loop relax. This process continues stepwise until the last midgut loop enters the abdominal cavity. A similar length of the SMA branches is a precondition for this sequence, as otherwise a disordered shift of random midgut loops could occur, which has not been observed in our study. However, we could not measure the length of the supplying vessels due to technical limitations. To conclude, the shift of the midgut involves continuous growth of the abdomen as the active part and stretched, rigid vessels of presumed similar length as the passive part which anchor the midgut and widen the umbilical ring.

However, the question of the transferability of findings in animal models to human embryology remains. For instance, it is known that the development of the yolk sac and its vessels differ in rats and humans, as these structures persist in rodent embryos[32–35]. This might raise the question if a persisting vitelline artery may interfere with our proposed mechanism of the midgut shift at ED 17. However, as described above, the tension on the SMA, which represents the central part of the vitelline artery, and its branches, is induced by a growing abdominal scope. The tension itself is located between the fixed aorta and the midgut loops which are held back by the narrow umbilical orifice.

Hence, a fixation of the SMA through a persisting peripheral rest of the vitelline artery would have no impact.

## Methods

**Study design**. This study is primarily based on morphological changes during embryogenesis. To provide a minimal statistic for measured values, a sample size of 4 animals per age group from ED12 onwards was considered sufficient. No data was excluded. The reproducibility is given, as two embryos per litter and two litters per age group were processed separately and investigated, if applicable (one exception is the shown return of the midgut loops from the umbilicus to the abdominal cavity). Additionally, age-matched embryos showed similar patterns and measured values. Randomization and blinding were not relevant to our study as only one group was investigated.

**Specimen**. Animal care and experimental procedures were approved by the institutional review board (state directorate Saxony, Referat 25, veterinary and food monitoring, Braustraße 2, 04107 Leipzig. Proposals: T14/15, T44/16, T13/18). This study neither involve wild animals nor samples collected from the field. To obtain rat embryos of defined gestational age, Sprague-Dawley rats were mated, and pregnancy was verified by the presence of a vaginal smear. Staging was performed by definition of the gestational age, with the day of positive vaginal smear defined as embryonic day 0 (ED 0). Animals were housed at the Medical Experimental Center of the University of Leipzig in rooms with a controlled temperature (22 °C), humidity (55%), and 12 h light–dark cycle. Food and water were supplied ad libitum. Pregnant rats were euthanized by an overdose of pentobarbital [300 mg/kg BW], and embryos were harvested.

**Sample preparation**. Overall, 50 embryos, regardless of their sex, aged from ED 10 to ED 21 were analyzed for the current study. Two dams per age group ($n_{total} = 24$) and at least two embryos per litter were used resulting in at least four embryos per age group, with the exception for ED 10 ($n = 1$) and ED 11 ($n = 3$). Additionally, ED 12 was subdivided into an early (early ED 12) and a late (late ED 12) stage resulting in four more embryos. For ED 17, eight embryos were analyzed. Each embryo was weighed, measured and fixed in Bouin's solution (9% formaldehyde, 5% acetic acid, and 0.9% picric acid) at RT. Fixation was performed for 3 days. Embryos of ED 10 – late ED 12 were left in the uterus, and older specimens were removed from the uterus. In embryos older than ED 18, the superficial abdominal skin layer was removed, improving the penetration of Bouin's solution. Subsequently, the samples were stored in 80% ethanol. To enhance image contrast, complete embryos were dehydrated using the "critical point" technique. Before drying, the samples were put in 100% ethanol for 24 h. Briefly, the alcohol was removed by washing several times with liquid carbon dioxide ($CO_2$) inside of a pressure chamber and subsequently dried utilizing the critical point dryer CPD 2 (Pelco, Ted Pella, Inc., CA, USA). All samples were stored at RT (Supplementary Fig. 8). The sample preparation (fixation in Bouin's solution and subsequent "critical point" drying) used in our study has been shown to be superior compared to, e.g., chemical or air drying as it preserves the surface structure of a specimen which could otherwise be damaged due to surface tension when changing from the liquid to gaseous state[35]. However, we did observe a systemic volumetric shrinkage of the embryos. Nevertheless, all structures remained preserved and no artifacts, such as fissures or breakages, had been found in the investigated structures.

**Micro-CT scanning**. Each rat embryo was analyzed using SkyScan 1172-100-50 (Bruker microCT, Kontich, Belgium). All samples were scanned with 40 kV and 250 μA without filter. The voxel size ranged from 2.04 to 7.63 μm, depending on the specimen size. Images were reconstructed with the scanner software (NRecon 1.7.0.4; Bruker) and converted to a bitmap-file-format.

**Segmentation**. The segmentation of embryonic structures was performed by CT-Analyzer (CTAn®, Version 1.16.1.0; Bruker). The structures were manually segmented by generating a series of regions of interest (ROIs) around the embryonic structure to extract the information (Supplementary Fig. 8).

**Statistics and reproducibility**. After data segmentation (CT Analyzer, Bruker microCT), the 3D viewing software CTvox® (Bruker microCT) was used to produce volume rendering and virtual sections for graphical illustrations and videos. The videos of the developing intestine (Supplementary videos 3 and 4) were generated using single pictures of different developmental stages. These pictures were then artificially interpolated to a morphing sequence using the free software FotoMorph (Version 13.9.1, digital photo software, http://www.diphso.no/FotoMorph.html).

Results are expressed as single data points ± standard deviation (SD). For comparison of length, volume, scope, and area measurements over time, one-way analysis of variance (ANOVA) with Bonferroni post-hoc tests was used. The effect sizes for multiple regressions are shown as Cohen's $f$. Graphs were designed with GraphPad Prism (La Jolla, CA, USA), $p$-values were calculated with the software SPSS (Version 26, IBM®, Armonk, NY, USA) and considered significant when $< 0.05$.

**Reporting summary**. Further information on research design is available in the Nature Research Reporting Summary linked to this article.

## Data availability

The micro-CT datasets consist of single images in.png format. The images in each set can be combined into three-dimensional structures by using appropriate software such as CTvox, CTAn (both software from Bruker, Kontich, Belgium), or Amira (software from Thermo Fisher Scientific, Waltham, MA, USA). The individual images can also be viewed with any graphic program. Our datasets and all supplementary videos are openly available in Publissio ZB MED Information Centre of Life Science at https://doi.org/10.4126/FRL01-006424446. All measured raw data is present in the Supplementary Data 1.

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

## Acknowledgements

We acknowledge Joerg Maenner (Institute of Anatomy and Embryology UMG, Georg-August-University Goettingen, Goettingen, Germany) for his assistance and critical comments on the paper.

## Author contributions

M.G., I.M., and D.K. designed the study. R.H. and M.L. provided experimental tools and equipment. The SkyScan 1172-100-50 (micro-CT), technical support, and supervision was provided by R.H. Manual segmentation of embryonic structures was done by M.G. and I.M. Interpretation of data and initial draft of the manuscript was done by M.G., I.M., M.L., and D.K. All authors contributed to the final manuscript editing.

## Funding

## Competing interests

The authors declare no competing interests.
