## [Peer Review File · Communications Biology]

Reviewers' comments:

Reviewer #1 (Remarks to the Author):

This is an impressive study following the dynamics of GIT development.
I have a number of questions:

How many dams were used per pregnancy day and how many embryos were analysed per litter?

Does sex of the embryo make a difference?

What is the impact of fixation?

Can they validate the major findings from in vivo imaging or freshly excised embryos?

The Video showing animated development from day 12 to 13 is not clear. How was it generated using fixed embryos?

Reviewer #2 (Remarks to the Author):

Ginzel and colleagues provide a “new” perspective to midgut development in rat embryos. The authors add 5 superb videos that show rat embryos in situ at ED11 and ED12, and the growth and initial looping of the midgut during ED12 and ED13. There is a combined Results and Discussion section. The text is a solid account of gut herniation and its return. After reading it, I was, nevertheless, a bit disappointed, since the account offers no novel insights or new mechanisms, possibly with the exception of the separation of the umbilical and extraembryonic coelom. The authors do introduce several new names for existing concepts, which is acceptable if they are properly discussed. Instead, the authors cite (relatively few) other authors, often quite literally, but hardly ever interpret these earlier findings and judge whether they do or do not fit in their model and reasoning. Examples underscoring these criticisms are given in the next paragraphs.

Concepts:

The pre-/intermediate/definitive umbilical ring (border of body wall) and the umbilical and vitelline funnels (tissues surrounding the umbilical and vitelline vessels) are not really new concepts. Anyone who has ever studied a mammalian embryo knows that the umbilical veins initially mark the ventral end of the lateral body wall (see e.g. Supplemental Fig 3). The vitelline and umbilical funnels are remarkably similar to drawings present in Hartwig et al. (*Am J Clin Pathol* 96: 1991; 640–647 (cited in ref 4, but not in the present manuscript)). The constant three intestinal loops in “semi-random loop arrangement” in the umbilical hernia are quite similar to the three “secondary loops” of Soffers et al (ref 7).

Lack of discussion:

Unfortunately, there is no real discussion of putative new findings. In lines 221-236, the initial looping of the midgut is described to include a change in position of the caecum. The authors explain this finding in their Conclusion as due to the convergence of the duodenal and colonic limb. Others have defined this change in position as a rotation. Although I do agree with the authors that this change in position does not represent a rotation, a convergence of the limbs of the loops alone does

not suffice to produce a positional change of the caecum relative to the duodenum with “sidedness”. Such an asymmetry in the loop requires a symmetry breaking event.

In lines 254ff, the authors argue that the attachment of the vitelline artery to the midgut wall causes gut herniation by “pulling” the loop out. Although an interesting hypothesis, this case should be weighed against other hypotheses. The authors argue that a too small relative size of the abdominal cavity cannot be a cause for herniation. I think their argument is reasonable. However, they do not even discuss the view of Ueda et al (ref 15), who claim differential growth as a cause. The already mentioned subloops of the gut in the umbilical hernia are not so different from the secondary loops of Soffers et al. (ref 7), but the difference is again not discussed.

In lines 341-2, the authors hypothesize that the return of the midgut into the abdominal cavity is mediated by an increasing stretch on the vitelline artery. Again, although the role of the vitelline artery in herniation and now return is a provocative hypothesis, little evidence in favor is produced. Then, in lines 347-8, the authors state that the shift was highly organized and took place in a clear order. That is unexpected for an elastic band as a driving structure. In lines 349-351 the authors have, indeed, to claim that each of the branches of the SMA features a distinct depletion of elasticity. Although I do not want to argue that this explanation is impossible, not a thread of evidence in favor is supplied. I would have expected that the authors would weigh their arguments against those of their ref 8 (Nagata et al), who claim that the rapid increase in intra-abdominal volume is the driving force. Nagata et al. supported their claim with measurements of the abdominal cavity. The authors then report on the sequence of the return of the respective parts of the midgut (lines 352-360), writing that the ileum is the last loop to enter the abdominal cavity. Unfortunately, they again do not discuss the literature as this sequence was already shown in human embryos by many groups (Snyder & Chaffin, Anat Rec 1954; 113: 451-7, not cited); Soffers et al, (ref 7); Nagata et al. (ref 8 and Congenit Anom 2020; 60: 87–8). None of these sources are discussed.

In summary, this study provides a solid and detailed description of midgut herniation and return, as well as some really beautiful material for the classroom, but does fail a balanced discussion of data and concepts with earlier studies. Since the Results, as argued, hardly contain discussion, a separate Discussion section improve clarity.

Details:

- There is no information on how the videos showing growth and initial looping of the midgut were made.
- The addition “high-resolution” to the μ CT technique used (line 422) applies to that technique, not to the images produced. Although voxels are three-dimensional, the authors only provide a one-dimensional size. Assuming that number reflects the length of one of the sides of the (isotropic?) voxel, the resolution approximates 20-50% of a cell diameter.
- The colonic flexure is mentioned in the text (line 231), but is not shown in one of the Figures. Supplemental Figure 6 would have been a good opportunity.
- Lines 193-4: The umbilical vein disintegrated completely into smaller vessels at late ED 12. Thus, a scaffold for the further development of the right lateral body wall had formed. How does a scaffold form from disintegrated vessels?
- The exponential growth of the intestines allows calculation of the daily growth rates. Such numbers are easier to compare than exponential lines.

Response to the comments of the Reviewers

We thank the Reviewers for carefully reading our manuscript and providing valuable comments and suggestions. The following changes to the manuscript have been made.

Reviewer #1:

We thank Reviewer #1 for his comments and thoughtful questions, especially for pointing out missing descriptions in the methods section.

Response 1)

“How many dams were used per pregnancy day and how many embryos were analyzed per litter?”

We thank the Reviewer for this question. The methods section was amended as follows: “Overall, 50 embryos regardless of their sex aged from ED 10 to ED 21 were analyzed for the current study. Two dams per age group ($n_{\text{total}}=22$) and at least two embryos per litter were used resulting in at least four embryos per age group, with the exception for ED 10 ($n=1$) and ED 11 ($n=3$). Additionally, ED 12 was subdivided into an early (early ED 12) and a late (late ED 12) stage resulting in four more embryos. For ED 17, eight embryos were analyzed.”

Response 2)

“Does sex of the embryo make a difference?”

The gender of rat embryos was not analyzed for the following reasons: First, in the early fetal period until ED 15, a clear distinction of the sex from morphologic details of the genital anlage is not possible (*Figel et al., 2011*); second, because of the low number of embryos (four embryos per age group) it was not possible to perform a statistical analysis. However, the methods section was amended as follows: “Overall, 50 embryos, regardless of their sex, aged from ED 10 to ED 21 were analyzed for the current study.”

Response 3)

“What is the impact of fixation?”

We thank the Reviewer for this question. We added the following paragraph to the methods: “The sample preparation (fixation in Bouin’s solution and subsequent “critical point” drying) used in our study has been shown to be superior compared to e.g., chemical or air drying as it preserves the surface structure of a specimen which could otherwise be damaged due to surface tension when changing from the liquid to gaseous state. However, we did observe a systemic volumetric shrinkage of the embryos. Nevertheless, all structures remained preserved and no artifacts had been found such as fissures or breakages.”

Response 4)

“Can they validate the major findings from in vivo imaging or freshly excised embryos?”

We thank the Reviewer for this suggestion. It is not possible to apply the microCT technique to living embryos because this technique requires an absolutely fixed specimen, as every movement (such as heart beat or breathing) would distort the scan. Furthermore, the exposure of living embryos to the high radiation of the microCT is ethically questionable. For the case of freshly excised embryos, a dissection of the abdominal cavity is required, which could result in artefacts. This was initially the main reason to conduct the study using the non-destructive microCT technique.

Response 5)

“The Video showing animated development from day 12 to 13 is not clear. How was it generated using fixed embryos?”

We thank the Reviewer for this remark and apologize for the missing description of the used animation technique. In short, we used single pictures of different developmental stages of midgut development. The single pictures were animated into a video using morphing

software. Therefore, we added the following text to the method section: “After data segmentation (CT Analyzer, Bruker microCT), the 3D viewing software CTvox® (Bruker microCT) was used to produce volume rendering and virtual sections for graphical illustrations and videos. The videos of the developing intestine (Supplemental videos 3 and 4) were generated using single pictures of different developmental stages. These pictures were then artificially interpolated to a morphing sequence using the free software FotoMorph (Version 13.9.1, digital photo software, <http://www.diphso.no/FotoMorph.html>).”

Reviewer #2:

We want to thank Reviewer #2 for his detailed remarks and suggestions on our manuscript. The main suggestion has been to create a separate discussion section in order to reflect our results in the light of relevant literature. In our initial submission, we intended to describe the observations made in form of an atlas. However, we believe that the introduction of a separate discussion has the potential to increase the impact of our manuscript. Hence, a discussion section was added, highlighting the following steps in midgut development: 1. Development of the umbilicus. 2. The development of the first midgut loop and its relationship to vitelline structures. 3. Cluster formation and staged shift of the midgut. In these sections, we addressed and discussed the points mentioned by the Reviewers. Consequently, the conclusion section has been removed. Furthermore, we removed the phrase “a new perspective” from the title of the manuscript.

Response “Concepts”

We divided the comment “concepts” into 3 questions [Q1-3].

“The pre-/intermediate/definitive umbilical ring (border of body wall) and the umbilical and vitelline funnels (tissues surrounding the umbilical and vitelline vessels) are not really new concepts. Anyone who has ever studied a mammalian embryo knows that the umbilical veins initially mark the ventral end of the lateral body wall (see e.g.

Supplemental Fig 3) [Q1]. The vitelline and umbilical funnels are remarkably similar to drawings present in Hartwig et al. [Q2]. The constant three intestinal loops in “semi-random loop arrangement” in the umbilical hernia are quite similar to the three “secondary loops” of Soffers et al (ref 7) [Q3].”

A1: We agree with the Reviewer, that the overall patterns of umbilical development are known. However, certain aspects we observed in our study may be helpful for the detailed understanding of umbilical development. We noticed, the importance of the umbilical vessels and its changes over time for the formation of definitive umbilicus. Therefore, we discuss umbilical development using the vessels as guiding structures: No vessels = Preumbilical ring; ring of the paired UVs = intermediate umbilical ring, reduction of the right UV and repositioning of the left UV = definitive umbilical ring. We believe, that this is a new view on the development of the umbilicus. To clarify certain aspects of vessel development, we added the following passage to our results: “During further development, the right umbilical vein, which formed the edge of the lateral body wall until early ED 12, disintegrated completely into smaller vessels at late ED 12 (Fig. 2c). The development of the left umbilical vein differed from the right side, by increasing in diameter. Simultaneously, through the complete disintegration of the right umbilical vein, the left umbilical vein loses its connection to the right vein and thus gets more and more separated from the lateral and caudal body wall. As a result, the left umbilical vein becomes an integral part of the definitive umbilicus at ED 13 (Fig. 2b, c).”

Additionally, this paragraph is a subject in our new discussion section.

A2: In our study, we came across structures which were the result of the complex process of folding of the embryo. These funnel like structures were formed by the twisted amnion, surrounding the umbilical and vitelline vessels. Comparing these structures to drawings published by *Hartwig et al.* and others, we noticed that in these illustrations either the amniotic membrane was removed from the embryo (*Hartwig et al.*) or showing only one

amniotic compartment (*Muniraman, H. et al.*). We addressed this point in the discussion section.

A3: We thank the Reviewer for his remark. The secondary loops of Soffers et al. and their differences to our described clusters are now discussed in detail.

“These clusters resemble basically the previously described 2nd to 4th secondary loops by Soffers et al.⁷. However, the subdivision by Soffers et al. is based on intestinal loop organization in a chronological order, while the three extraembryonic secondary loops are connected to only two mesenteric leafs. Our classification uses vessel morphology, which allows a clear identification of each cluster, based on the three bundles of vessels. Furthermore, we do not subdivide these clusters further into tertiary and quarterly loops, as proposed by Soffers et al. because this subdivision was not supported by the observed vessel pattern.“

Lack of discussion:

Q1: “Unfortunately, there is no real discussion of putative new findings. In lines 221-236, the initial looping of the midgut is described to include a change in position of the caecum. The authors explain this finding in their Conclusion as due to the convergence of the duodenal and colonic limb. Others have defined this change in position as a rotation. Although I do agree with the authors that this change in position does not represent a rotation, a convergence of the limbs of the loops alone does not suffice to produce a positional change of the caecum relative to the duodenum with “sidedness”. Such an asymmetry in the loop requires a symmetry breaking event.”

A: We thank the Reviewer for his remarks and suggestions. However, we feel that in the remarks made, the colonic flexure might better fit to the discussed morphological changes than the caecum. We addressed this aspect in the new discussion section. We believe, that

the mechanism behind the side-by-side movement is well explained by *Ueda et al.* and *Metzger et al.*

Q2: “In lines 254ff, the authors argue that the attachment of the vitelline artery to the midgut wall causes gut herniation by “pulling” the loop out. Although an interesting hypothesis, this case should be weighed against other hypotheses.

A: We thank the Reviewer for his suggestion and agree that the term “pulling of the midgut loop” is easy to misunderstand and does not describe the process in sufficient detail. Therefore, we now addressed this topic in detail in the discussion and added an explanatory scheme to the supplemental figures (Supplemental Fig. 7).

Q3: “The authors argue that a too small relative size of the abdominal cavity cannot be a cause for herniation. I think their argument is reasonable. However, they do not even discuss the view of Ueda et al (ref 15), who claim differential growth as a cause. The already mentioned subloops of the gut in the umbilical hernia are not so different from the secondary loops of Soffers et al. (ref 7), but the difference is again not discussed.”

A: We thank the Reviewer for his recommendation. Mall and others tried to explain PUH formation by a relatively small size of the abdominal cavity in relation to the growing midgut. We discussed this problem in more detail by introducing a recent study published by *Kanahashi et al.* However, we feel, that *Ueda et al.* do not describe the initial process of PUH formation, but rather processes involved in the first midgut rotation. We discuss this topic and furthermore the secondary loops of *Soffers et al.* and compare them to the clusters described in our study in detail.

Q4: “In lines 341-2, the authors hypothesize that the return of the midgut into the abdominal cavity is mediated by an increasing stretch on the vitelline artery. Again, although the role of the vitelline artery in herniation and now return is a provocative hypothesis, little evidence in favor is produced.”

A: We thank the Reviewer for pointing out the missing evidence. We therefore analyzed the diameters of the supplying blood vessels, arising from the SMA to the midgut loops before, during and after the midgut shift. A new supplemental figure (Supplemental Fig. 4) and a new paragraph was added to the result section:

“To assess the assumed stretching of the supplying midgut vessels we analyzed the diameters of the SMA and its branches. In addition, we measured the volume of the SMA to calculate its length. Our morphometric data showed that the diameter of the SMA was smaller dorsally compared to ventral. The calculated length of the SMA increased from ED 16 to ED 17.0 and decreased significantly after the shift (Figure 5c). The mean diameter of the SMA branches showed a stagnation from ED 16 to ED 17 and a significant increase after the shift (Supplemental Fig. 4).”

These results are also discussed in the new discussion section.

Q5: “Then, in lines 347-8, the authors state that the shift was highly organized and took place in a clear order. That is unexpected for an elastic band as a driving structure. In lines 349-351 the authors have, indeed, to claim that each of the branches of the SMA features a distinct depletion of elasticity. Although I do not want to argue that this explanation is impossible, not a thread of evidence in favor is supplied. I would have expected that the authors would weigh their arguments against those of their ref 8 (Nagata et al), who claim that the rapid increase in intra-abdominal volume is the driving force. Nagata et al. supported their claim with measurements of the abdominal cavity. The authors then report on the sequence of the return of the respective parts of the midgut (lines 352-360), writing that the ileum is the last loop to enter the abdominal cavity. Unfortunately, they again do not discuss the literature as this sequence was already shown in human embryos by many groups (Snyder & Chaffin, Anat Rec 1954; 113: 451-7, not cited); Soffers et al, (ref 7); Nagata et al. (ref 8 and Congenit Anom 2020; 60: 87–8). None of these sources are discussed.”

A: We thank the Reviewer for these suggestions. We discussed all points of this comment in our discussion section in detail: “This shift itself was first described by *Mall* and later by *Frazier & Robbins*. *Mall* proposed in 1899, that the midgut loops inside the extraembryonic coelom gets “sucked back” into the abdominal cavity. He suggested, that an extension of the abdominal cavity resulted in a kind of negative pressure. In 1915, *Frazier & Robbins* modified the mechanism for the shift, indicating that amniotic pressure on the umbilical sac might contribute to the “suck back” mechanism. The shift itself is described to take place in a specific order; the intestinal segments shift from proximal to distal and not en block, as a narrow umbilical orifice would make this process unlikely. Based on results from *Soffers et al.*, *Nagata et al.* proposed a mechanism, called “wrapped model”. In short, this theory describes a process in which the intestinal loops are wrapped into the abdominal cavity as the height of the umbilical ring increases ventrally.

Interpreting our results, we propose the following mechanism: The shift is triggered by the increasing abdominal scope. This leads to a ventral push on the umbilical ring. As the umbilical orifice is too small at this stage the midgut loops inside the extraembryonic coelom get pushed consequently to ventral. This results in an increased tension on the SMA, which is anchored to the aorta, and subsequent on its branches to the midgut loops. We assume that this stretch causes a loss of vessel elasticity, hence keeping the loops in a fixed distance to the aorta. Additionally, the stretched branches align to a cone like structure, which widens the umbilical orifice shortly before the shift takes place (Supplemental Fig. 3). Due to the morphology of the SMA and its branches (spiral staircase) the shift of the midgut takes place in the previous described organized fashion, from dorsal to ventral, loop by loop, following the vascular outlets of the supplying vessels. This means, that the shift of the midgut is a dynamic process which involves continuous growth of the abdomen, anchoring of midgut loops by stretched vessels, and widening of the umbilical ring. This process does not include active pull or contraction of the midgut and/or vessels and is therefore passive. The fact, that a stretch of vessels is involved is shown by their relaxation after the shift.”

Q6: “In summary, this study provides a solid and detailed description of midgut herniation and return, as well as some really beautiful material for the classroom, but does fail a balanced discussion of data and concepts with earlier studies. Since the Results, as argued, hardly contain discussion, a separate Discussion section improve clarity.”

We thank the Reviewer for this positive summery. A separate discussion section was added accordingly.

Details:

Q1: “There is no information on how the videos showing growth and initial looping of the midgut were made.”

A: We thank the Reviewer for his remark and apologize for the missing description of the used animation technique. In short, we used single pictures of different developmental stages of midgut development. The single pictures were animated into a video using morphing software. Therefore, we added the following text to the method section: “After data segmentation (CT Analyzer, Bruker microCT), the 3D viewing software CTvox® (Bruker microCT) was used to produce volume rendering and virtual sections for graphical illustrations and videos. The video of the developing intestine was generated using single pictures of different developmental stages. These pictures were then artificially interpolated to a morphing sequence using the free software FotoMorph (Version 13.9.1, digital photo software, <http://www.diphso.no/FotoMorph.html>).”

Q2: “The addition “high-resolution” to the μ CT technique used (line 422) applies to that technique, not to the images produced. Although voxels are three-dimensional, the authors only provide a one-dimensional size. Assuming that number reflects the length of one of the sides of the (isotropic?) voxel, the resolution approximates 20-50% of a cell diameter.”

A: We thank the Reviewer for that remark. We thus removed the term “high resolution” throughout the manuscript.

Q3: “The colonic flexure is mentioned in the text (line 231), but is not shown in one of the Figures. Supplemental Figure 6 would have been a good opportunity.”

We thank the Reviewer for this remark, but we believe that the colonic flexure, mentioned in the text (line 231), is shown in different developmental stages in figure 4a. This flexure, also visible in supplemental Figure 6, shows a later developmental stage and is therefore not mentioned in (previous) line 231 of the manuscript.

Q4: “Lines 193-4: The umbilical vein disintegrated completely into smaller vessels at late ED 12. Thus, a scaffold for the further development of the right lateral body wall had formed. How does a scaffold form from disintegrated vessels?”

We thank the author for this comment and agree that the chosen term “scaffold” is misleading. We removed this term and specified the vessel development as shown in “response 1” to Reviewer #2.

Q5: “The exponential growth of the intestines allows calculation of the daily growth rates. Such numbers are easier to compare than exponential lines.”

We thank the Reviewer for this suggestion. A table with all collected data was added to the supplemental section.

REVIEWERS' COMMENTS:

Reviewer #2 (Remarks to the Author):

My comments were adequately addressed. I have no further comments

Reviewer #3 (Remarks to the Author):

Ginzel and coworkers have profoundly revised their manuscript. It now provides more focus. However, the one aspect that remains underexposed is the prominent species difference with respect to the yolk sac and vitelline artery between men and rats. Rodents have a persisting yolk sac, a temporary yolk sac placenta, and a correspondingly large vitelline artery, whereas humans do not. In my opinion, the authors should discuss the potential effect(s) of this species difference, which has a major effect on the size of the vitelline artery, on their conclusions.

- The authors emphasize the role of the vitelline artery in the return/shift of the hernia. That is an interesting and new mechanistic hypothesis. Somewhat surprisingly, they do not discuss the potential role(s) of the persistence of the yolk sac in rodents, and the correspondingly wide vitelline artery and vein. By the time the intestines return, the vitelline artery and vein have disappeared in humans. Of course, the part of the vitelline artery that perfuses the guts remains, but the size of the stem is surely different in rats from that in men.

- The rodent embryo undergoes a pronounced turning of its body axis, whereas human embryos do not or hardly experience such a turning. The turning is accompanied by a twist of the body axis and, consequently, by a pronounced (head-)fold in their amnion, which in turn, accounts for the shape of the vitelline compartment. An important body of the Results and Discussion sections are devoted to this structure even though human and rat herniation and return are similar. The question, therefore, arises, how important this amniotic headfold is.

- The pre-, intermediate and definitive umbilical rings are introduced as an important modification of Hartwig's model. This boils down to the perennial issue of splitting versus lumping. In this case, even the description of constituting tissues of the pre- and definitive umbilical rings are the same, only the age of the embryo differs. The authors' descriptions are: "The embryo's membranes were in continuity with the embryonic ectoderm and mesoderm of the lateral body wall of the embryo" at ED11.5. At ED13, "The umbilical ring was no longer formed by the umbilical vessels. Instead, the developing ventrolateral body walls (somatopleura) now represented the boundaries of the definitive umbilical ring." We all know, of course, that the right umbilical vein regresses and disappears, and that this a process without pronounced interruptions, so why would we need stages if they do not have a functional consequence. My interpretation of the text is that the umbilical ring only plays a role during the return of the intestines, so long after the pre- and intermediate umbilical stages have passed.

- I do not quite understand the reason why the colon acquires a left-cranial position. The authors attribute it to the right-sided position of the duodenum in the abdomen. The obvious questions are how the duodenum got its right-sided position, and if this right-sided position developed earlier than the colonic left-sided position. The reason to raise this point is that it could well be that the position of the duodenum and colon are mediated by the same symmetry-breaking process.

- What will probably be best remembered of this study is the proposed mechanism of the return. Thus far, the increased volume of the abdominal cavity was often postulated to account for sufficient suction. The stretching of the SMA branches is, therefore, an intriguing new proposal. The stretch amounts to ~10%/day, and the subsequent shortening also. Can that change mediate enough force to make the guts enter the abdominal cavity. In this respect it is also important that the branches of the SMA are sufficiently different in length that the returning loops do not get jammed at the entrance. Was this checked? From the reconstructions it looks like the main stem (so the persisting vitelline artery) was the widest, but that vessel remains attached to the yolk sac. And how did the colon piggy-back? Some additional clarification would be useful. In that discussion, some words could also be spent on whether a combination of effects could be responsible for the return?

- Does the intraabdominal space increase relative to overall body growth and intestinal growth, or does it increase disproportionately?

- The more recent accounts of gastroschisis state that is a defect in the right paraumbilical body wall due to an ischemic event (Rittler M, Vauthay L, Mazzitelli N. *Birth Defects Res A Clin Mol Teratol.* 2013;97(4):198-209; Bargy F, Beaudoin S. *Fetal Diagn Ther.* 2014;36(3):223-30.

Details:

I find the abbreviation "PUH" unfortunate, because the sequence of capitals has no direct relation to the "physiological herniation of the gut into the umbilicus" (Introduction).

The Discussion states that "previous reports present drawings, which suggest that the yolk sac and the umbilical cord are located in a single compartment surrounded by the amniotic membrane 4,17,22,24,25. What is wrong with this statement?

Why is ED 14 the last stereotypical morphological stage?

There is a typo in Figure 6: "Proportionon"

Response to the comments of the Reviewers

We thank the Reviewers for carefully reading our manuscript and providing valuable comments and suggestions. The following changes to the manuscript have been made.

Reviewer #3:

Q1: Ginzel and coworkers have profoundly revised their manuscript. It now provides more focus. However, the one aspect that remains underexposed is the prominent species difference with respect to the yolk sac and vitelline artery between men and rats. Rodents have a persisting yolk sac, a temporary yolk sac placenta, and a correspondingly large vitelline artery, whereas humans do not. In my opinion, the authors should discuss the potential effect(s) of this species difference, which has a major effect on the size of the vitelline artery, on their conclusions. The authors emphasize the role of the vitelline artery in the return/shift of the hernia. That is an interesting and new mechanistic hypothesis. Somewhat surprisingly, they do not discuss the potential role(s) of the persistence of the yolk sac in rodents, and the correspondingly wide vitelline artery and vein. By the time the intestines return, the vitelline artery and vein have disappeared in humans. Of course, the part of the vitelline artery that perfuses the guts remains, but the size of the stem is surely different in rats from that in men. The rodent embryo undergoes a pronounced turning of its body axis, whereas human embryos do not or hardly experience such a turning. The turning is accompanied by a twist of the body axis and, consequently, by a pronounced (head-)fold in their amnion, which in turn, accounts for the shape of the vitelline compartment. An important body of the Results and Discussion sections are devoted to this structure even though human and rat herniation and return are similar. The question, therefore, arises, how important this amniotic headfold is.

A1: We thank the Reviewer for this question. The discussion section was amended as follows: "However, the question of the transferability of findings in animal models to human embryology remains. For instance, it is known that the development of the yolk sac and its

vessels differ in rats and humans, as these structures persist in rodent embryos. This might raise the question if a persisting vitelline artery may interfere with our proposed mechanism of the midgut shift at ED 17. However, as described above, the tension on the SMA, which represents the central part of the vitelline artery, and its branches, is induced by a growing abdominal scope. The tension itself is located between the fixed aorta and the midgut loops which are held back by the narrow umbilical orifice. Hence, a fixation of the SMA through a persisting peripheral rest of the vitelline artery would have no impact.”

Q2: What will probably be best remembered of this study is the proposed mechanism of the return. Thus far, the increased volume of the abdominal cavity was often postulated to account for sufficient suction. The stretching of the SMA branches is, therefore, an intriguing new proposal. The stretch amounts to ~10%/day, and the subsequent shortening also. Can that change mediate enough force to make the guts enter the abdominal cavity. In this respect it is also important that the branches of the SMA are sufficiently different in length that the returning loops do not get jammed at the entrance. Was this checked? From the reconstructions it looks like the main stem (so the persisting vitelline artery) was the widest, but that vessel remains attached to the yolk sac. And how did the colon piggy-back? Some additional clarification would be useful. In that discussion, some words could also be spent on whether a combination of effects could be responsible for the return?

A2: We thank the Reviewer for this suggestion. The discussion section was specified as follows: “Interpreting our results, we propose the following mechanism: The shift is triggered by the increasing abdominal scope. In effect, this scope increases the distance between the aorta and the umbilical orifice. Consequently, the midgut loops inside the extraembryonic coelom are constantly pushed to ventral, as the umbilical orifice is too small for the shift to occur until ED 17. From ED 16 to ED 17 the SMA branches get stretched in dorso-ventral direction forming a cone-like structure which subsequently widens the umbilical orifice before the shift takes place (Supplemental Fig. 3). The shift itself takes place in a highly organized fashion. It starts with the proximal loops of the 1st cluster and ends with the most distal loops

of the 3rd cluster. The cause for this order is the arrangement of the supplying vessels to the midgut loops. Their outlets from the SMA are orientated in a dorso-ventral manner with the most dorsal SMA branches supplying the proximal midgut loop and the most ventral branches supplying the distal loop. In detail, during the ongoing ventral shift of the umbilical orifice and the stretching of all SMA branches, the supplying vessels to the most proximal midgut loop are the first to lose their elasticity and thus become rigid. When this happens, this loop shifts through the ventrally moving umbilical orifice into the abdominal cavity. Afterwards, the vessels supplying this loop relax. This process continues stepwise until the last midgut loop enters the abdominal cavity. A similar length of the SMA branches is a precondition for this sequence, as otherwise a disordered shift of random midgut loops could occur, which has not been observed in our study. However, we could not measure the length of the supplying vessels due to technical limitations. To conclude, the shift of the midgut involves continuous growth of the abdomen as the active part and stretched, rigid vessels of presumed similar length as the passive part which anchor the midgut and widen the umbilical ring.”

Q3: I do not quite understand the reason why the colon acquires a left-cranial position. The authors attribute it to the right-sided position of the duodenum in the abdomen. The obvious questions are how the duodenum got its right-sided position, and if this right-sided position developed earlier than the colonic left-sided position. The reason to raise this point is that it could well be that the position of the duodenum and colon are mediated by the same symmetry-breaking process.

A3: We assume that the one reason of the right-sided position of the duodenum is the remaining intraembryonic left vitelline vein, which might orientate the duodenum to the right. We specified this in the text as follows: “Due to the position of the left vitelline vein inside the embryo the duodenal flexure was located to the right side of the abdomen”

Q4: Does the intraabdominal space increase relative to overall body growth and intestinal growth, or does it increase disproportionately?

A4: We thank the Reviewer for this question. Although we did not specifically look after this question, in our observations we did not note any “free space” in the abdominal cavity, which could reasonably be the cause of the shift of the midgut into the abdomen.

Q5: The more recent accounts of gastroschisis state that it is a defect in the right paraumbilical body wall due to an ischemic event (Rittler M, Vauthay L, Mazzitelli N. *Birth Defects Res A Clin Mol Teratol.* 2013;97(4):198-209; Bargy F, Beaudoin S. *Fetal Diagn Ther.* 2014;36(3):223-30).

A5: We thank the Reviewer for this suggestion. In the recent paper *Beaudoin et al.* discussed in detail the etiology and embryology of gastroschisis. In this paper the author expressed her disbelief in the theory of an ischemic event in stating that “Nevertheless, it was dismissed by both experimental results and embryological considerations, as this artery does not in fact

supply the body wall, but the gut" (*Beaudoin S. Insights into the etiology and embryology of gastroschisis. Semin Pediatr Surg. 2018*). We agree with the author in this point.